# Sentiment Dimensions and Intentions in Scientific Analysis: Multilevel Classification in Text and Citations

**Aristotelis Kampatzis** *[ID]**, Antonis Sidiropoulos** *[ID]**, Konstantinos Diamantaras** [ID] **and Stefanos Ougiaroglou** [ID]

Department of Information and Electronic Engineering, International Hellenic University (IHU), 57400 Thessaloniki, Greece; kdiamant@ihu.gr (K.D.); stoug@ihu.gr (S.O.)
* Correspondence: kampatzistelis@gmail.com (A.K.); asidirop@ihu.gr (A.S.)

**Abstract:** Sentiment Analysis in text, especially text containing scientific citations, is an emerging research field with important applications in the research community. This review explores the field of sentiment analysis by focusing on the interpretation of citations, presenting a detailed description of techniques and methods ranging from lexicon-based approaches to Machine and Deep Learning models. The importance of understanding both the emotion and the intention behind citations is emphasized, reflecting their critical role in scientific communication. In addition, this study presents the challenges faced by researchers (such as complex scientific terminology, multilingualism, and the abstract nature of scientific discourse), highlighting the need for specialized language processing techniques. Finally, future research directions include improving the quality of datasets as well as exploring architectures and models to improve the accuracy of sentiment detection.

**Keywords:** natural language processing (NLP); machine learning; deep learning; sentiment analysis; scientometrics; sentiment analysis of scientific citations





## 1. Introduction

Starting with the definition, sentiment analysis is a growing field of science that intersects with fields such as Artificial Intelligence (AI), Statistical Analysis (SA), and Natural Language Processing (NLP). Its central goal is to identify and evaluate the emotional expressions contained in texts. This approach uses various methods of data analysis to identify and evaluate the different nuances of the emotions and subjective elements expressed. Key work in this field includes the detection of emotion polarity (positive, negative, neutral), extraction of opinion elements, and overall emotional perception of texts [1].

In recent years, the problem of emotion analysis has attracted the interest of the scientific community, and the ability to assess people's preferences quickly and reliably for a topic has lead many companies and organizations to invest in this process. According to M. Wankhade et al. [1], applications of sentiment analysis are very useful in areas such as companies (product and service evaluations), the health sector for categorizing medical data, art (music, movie reviews, etc.), and social networks for monitoring public opinion. In addition, Sentiment Analysis has been explored at different levels, such as the Document Level, Sentence Level, Phrase Level and Aspect Level, as shown in Figure 1.

The Document Level focuses on evaluating the emotional charge of a whole text, with the purpose being to determine whether the document has positive, negative, or neutral emotional connotations. Both supervised and unsupervised learning approaches can be used. However, this type is not often used, mainly due to the large number of ideas and conflicting emotions. The Sentence Level focuses on assessing the emotion conveyed by each individual sentence. This method allows for a more detailed analysis compared to the Document Level, as it separates the text into sentences to evaluate the sentiment of each one individually. The Phrase Level focuses on specific expressions within a sentence and identifies the emotion present in smaller sections of the text. This level of analysis can reveal subtle variations in emotion that may be lost in a more generalized analysis at the

Document or Sentence Levels. Aspect Level analysis focuses on understanding the emotion associated with specific features of a product or service. For example, in mobile phone reviews, aspects may include design, durability, performance, battery life, camera, etc.

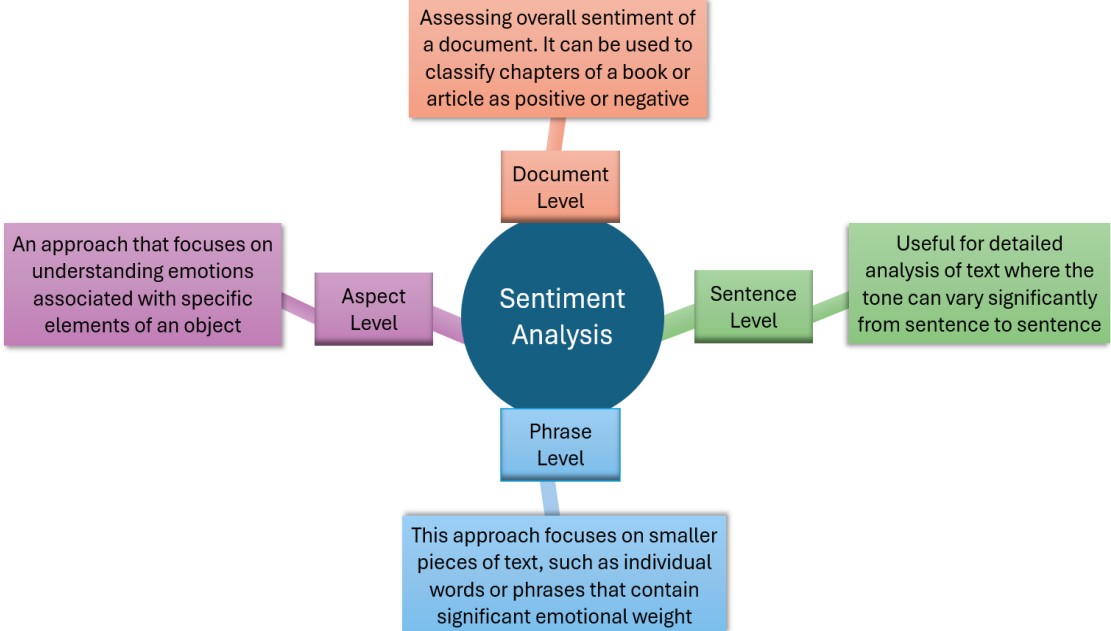

**Figure 1.** Sentiment Analysis Approaches.

The importance of literature references in the world of scientific research is a long-lasting and dynamic phenomenon. As the scientific community grows and evolves in the digital age, citations continue to be a vital link between research papers, allowing for interaction, acknowledgement, and critique between researchers. In this context, digital libraries and analytical services provide a rich source of information, facilitating access to, and evaluation of, scientific papers [2]. In fact, a citation is a textual element in a scientific publication that highlights and links to previous work for various reasons. It can be used to compare or highlight and identify different sources or previous work, thus contributing to academic discussion and scientific debate [2].

According to Alvarez et al. [3], in the field of citation analysis, qualitative evaluation is as important as quantitative evaluation, with the latter focusing on the frequency of citations. It is also argued that citations present different weights depending on the influence of the works that cite them, with it being thought that sentiment analysis can enhance the evaluation of the influence of scientific works by considering the author's disposition towards the cited work. Similarly, in [4,5], the authors provide a detailed examination of both quantitative and qualitative evaluation of citations. The quantitative evaluation concerns the frequency of citations and how this correlates with various aspects of the research work. On the other hand, the qualitative evaluation focuses on the quality of the citations, examining their importance, relevance, and weight within the text, and it is considered to be more critical than a quantitative evaluation. Therefore, by considering both quantity and quality, researchers can gain a more complete picture of both the influence and importance of a work in the scientific field.

Many research papers define a text that includes citations to a publication as a "citation context". They classify citations into being either explicit and implicit, with an explicit citation involving one or more sentences around a citation position in a document. This means that explicit citations are those that directly and clearly mention a source or previous work within the text of the article, usually stating the names of the authors. In contrast, an implicit or implied citation is a sentence that is not directly linked to the cited article and

is usually quoted within the text following an explicit citation [2,6,7]. For example, in the following text:

> *"Gregori et al. [19] introduced an innovative algorithm for sentiment analysis, leveraging a revolutionary methodology that enables the identification of nuanced emotional nuances within textual data. This state-of-the-art approach provides an adaptable, user-defined, and context-independent framework for sentiment analysis, thereby enhancing accuracy and efficiency in natural language processing tasks".*

The first sentence, "*Gregori et al. [19] introduced . . . within textual data*", is an explicit citation, while the second sentence, "*This state-of-the-art ... natural language processing tasks*", is an implicit citation.

Athar and Teufel [7] examine the detection of implicit citations in sentiment analyses of scientific texts. They emphasize the importance of including such citations to improve the quality of the overall polarity assignment. Finally, they point out the weakness of many recognition techniques, which usually ignore implicit citations by focusing only on citations that contain a direct reference to the author's name and publication date.

As the above demonstrates, the citation framework is an important resource for a variety of applications that need to identify the purpose or thematic objective of a citation, the reasons for citing a particular idea, as well as the critique of concepts that have preceded it in the academic literature. It is very important for new researchers to be able to understand the perspective of a project in a particular field; therefore, they will be able to discover any gaps in the literature if they identify a citation with a negative polarity or acknowledge the researchers' contribution by identifying citations with a positive sentiment [2]. The development of methods to evaluate citations with a deeper understanding and accuracy, focusing on both quality and quantity, has proven to be challenging. Sentiment analysis, as part of this approach, reveals new dimensions in evaluating the impact and contribution of a scientific project, thus helping to better understand the value of scientific communication as it impacts the academic community.

Recognizing that scientific texts hide a wealth of affective cues that are often overlooked, this study aims to provide a framework for analyzing these affective data. The aim of this review is therefore to highlight the importance of the emotional expressions that emerge in texts and scientific publications. The study aims to reveal patterns in the ways that emotions influence scientific discourse and the judgments that are formed around research results. Using modern Natural Language Processing and Neural Network techniques, it encourages the development of advanced systems capable of detecting and analyzing both the emotional connotations and the intensity of the reactions behind citations. The aim is to enhance transparency and accuracy in scientific communication, as well as to ensure a framework that encourages critical thinking and the constant review of research methods.

## 2. Research Methodology

The continuous development and evolution of research in each field makes it necessary to carry out extensive literature reviews to summarize and evaluate existing knowledge. In this context, previous work in the field of NLP and Sentiment Analysis has focused on the analysis of specific areas and other specific subject areas while also remaining limited in terms of methodology and scope. In contrast, this paper aims to provide a broader and more systematic literature review using the PRISMA (Preferred Reporting Items for Systematic Reviews and Meta-Analyses) methodology [8]. The PRISMA methodology, which is based on rigorous criteria for selecting and evaluating items, allows for the development of a transparent and reproducible literature review, thus providing significant added value to the field. Therefore, to conduct our systematic review, we followed the below steps:

- Defining the research questions.
- Searching for literature in reliable repositories.
- Setting criteria for rejecting certain papers.

- Removal of duplicate documents.

## 2.1. Research Questions

Below are the research questions that will be addressed in this study in order to explain the importance of classification in texts. Through these questions, we will examine how classification can contribute to exploring the role of each citation, highlighting the complexity of scientific discourse. In addition, these questions will seek to highlight the challenges faced by Sentiment Analysis while also exploring the contribution of advanced Machine Learning techniques that improve the evaluation of scientific research.

- *RQ1*. What algorithms and models have been developed for Sentiment Analysis in texts and how do they compare with traditional methods?
- *RQ2*. What preprocessing methods and classification accuracy metrics are applied in Sentiment Analysis?
- *RQ3*. In which cases do Machine Learning models perform better compared to Deep Learning models?
- *RQ4*. Which types of learning are most often used in classification problems in Sentiment Analysis?
- *RQ5*. How can Sentiment Analysis improve the understanding and evaluation of scientific communication?
- *RQ6*. What are the challenges in Sentiment Analysis in scientific texts?
- *RQ7*. What classifications are generally applied in the analysis of reporting frameworks?
- *RQ8*. Are there datasets available for Sentiment Analysis in citation contexts?
- *RQ9*. What is the role of emotions in communicating scientific results and how do they affect the acceptance of information?

All research questions will be answered in Section 5 (Discussion) after presenting the literature review.

## 2.2. Search Strategy and Selection Criteria

To find articles covering Sentiment Analysis in text and citations, we selected eight (8) databases: Springer, Google Scholar, Semantic Scholar, Science Direct, Association for Computing Machinery (ACM), MDPI, ACL Anthology, and IEEE Xplore. In each database, we performed several search queries to identify articles related to our review topic. The search queries were defined based on the requirements of each database, selecting and combining keywords to match the scientific and research focus of each platform. In general, we did not apply strict temporal search filters. In some of the queries, there was a need to restrict results, resulting in us activating a filter for the year of publication. Table 1 lists the queries that returned the most relevant results. In some platforms, however, it took more than one query before we found results that covered the scope of our work, while in others, such as IEEE, we identified relevant results with just one query. We also identified criteria for including and excluding articles in order to focus on the topic of the review. The included papers were screened to meet the following selection criteria:

- Be Conference Papers or Journal Articles.
- Apply NLP and Machine Learning methods.
- Apply Sentiment Analysis methods in citation contexts.
- Be Research Papers.
- The full text is available.
- Be published in reputable Journals or Conferences that show high-quality research.

Additional reasons for rejecting articles are as follows:

- *Rejection due to contradictions*. If there are contradictions in the data or results presented, the article may not be credible.
- *Rejection based on content*. If the screening process finds that the content of the article is not relevant to the topic of our study, we reject it.

**Table 1.** Search queries.

| Digital Repositories/Databases | Number of Query | Query |
|---|---|---|
| Springer | 1 | with the exact phrase: Sentiment Analysis Challenges. with at least one of the words: sentiment analysis challenges methods [Filters] year: 2021–2022 |
| | 2 | with at least one of the words: Scientometrics Citation. where the title contains: "Citation Context" OR "Citation Function Classification" |
| | 3 | with at least one of the words: Polarity Classification. where the title contains: "Polarity Classification" AND "Twitter" |
| | 4 | with all the words: Automatic Content Extraction. with the exact phrase: Named-entity Recognition. with at least one of the words: Sentiment Analysis Polarity Detection. where the title contains: "Sentiment Analysis" AND "Mining" [Filters] year: 2014–2019 |
| | 5 | with at least one of the words: Scientific Citation Sentiment Function BERT. where the title contains: "Scientific Citations" OR "BERT" AND "Formal Citation" |
| Google Scholar | 1 | ("sentiment analysis" AND "emotions") AND ("Word2Vec") AND "lexicon" AND ("word embeddings") AND "NLP" AND "machine learning" AND "online user reviews" |
| | 2 | ("Text Classification" AND "Product Reviews") AND ("Sentiment Analysis" OR ("Support Vector Machines" AND "TF-IDF" AND "Naive Bayes" AND "BERT") |
| | 3 | "sentiment classification" AND "comparative experiments" AND "product reviews" OR "text reviews" |
| | 4 | "Patterns" AND "Scientometrics" AND "Scientometrics Analysis" AND "Citation Analysis" |
| | 5 | "Sentiment Analysis" AND "Natural Language Toolkit" AND ("Twitter Messages" OR "tweets") AND "Word2Vec" AND ("CBOW" AND "Skip-Gram") |
| | 6 | "Sentiment Analysis" OR "Scientometric Analysis" AND "Convolutional Neural Networks" AND "CNN" AND "KNN" AND "Explicit Features" |
| | 7 | "Scientometrics" AND "citation function" AND "citation role" |
| | 8 | "Role" AND "Negative Citations" AND "natural language processing" AND "objective citations" |
| | 9 | Bibliometric AND "Analysis Methods" AND PageRank AND "Author citation" |
| | 10 | "Conditional random fields" AND "Extracting citation metadata" AND "citation indexing" AND "CiteSeer" AND "Extracting Citation Contexts" |
| | 11 | "BERT" AND "Attention Layer" AND "Sentiment Classification" AND "Attention" AND "Classification" AND "Citation" AND "Dictionary" |

**Table 1.** *Cont.*

| Digital Repositories/Databases | Number of Query | Query |
|---|---|---|
| Semantic Scholar | 1 | "Basic Emotions" AND "Detection of Implicit Citations" [Filters] Fields of Study: Psychology, Computer Science Date Range: 1990–2012, Has PDF = ON |
| | 2 | "Characteristics" AND "Citing Paper" AND "Cited and Citing" [Filters] Fields of Study: Computer Science Date Range: 1980–2007, Has PDF = ON |
| | 3 | "citation identification" AND "text citations" AND "Citation sentiment analysis" AND "Analysis Using Word2vec" AND "CBOW" OR "Skip-Gram" [Filters] Fields of Study: Computer Science, Has PDF = ON |
| Science Direct | 1 | ("Sentiment Analysis" AND "word embeddings" AND "Machine Learning") AND ("Sentiment lexicon" OR emotions OR "lexicon-based") AND "Supervised Machine Learning" |
| | 2 | ("Sentiment Analysis" AND "Reviews") AND ("LSTM" OR "Word2vec" AND ("RNN" OR "CNN") AND ("CBOW" OR "Skip-gram") |
| Association for Computing Machinery (ACM) | 1 | [[[Full Text: tweets] AND [Full Text: hashtags]] OR [[Full Text: "hashtag sentiment"] AND [Full Text: "sentiment lexicon"]]] AND [Title: tweets hashtags] AND [[Title: sentiment] OR [Title: lexicon]] |
| | 2 | [All: "citation recommendation system"] AND [All: "citation recommendation"] |
| MDPI | 1 | (Title: Sentiment Analysis) AND (Title: Social Media) OR (Title: Scientometric Analysis) AND (Title: Convolutional Neural Networks) AND (Full Text: CNN) OR (Full Text: NER) [Filters] year: 2021–2022, Journals: Electronics and Information, Article Types: Article |
| ACL Anthology | 1 | "Sentiment Detection" AND "Polarity" AND "Citation" AND "Implicit Citations" OR "Survey in Sentiment" |
| | 2 | "HMM" AND "Hidden Markov Models" AND "CRF" AND "Conditional Random Fields" AND "Information Extraction" |
| | 3 | Dataset Bibliographic Research |
| | 4 | Citation Analysis AND Neural networks |
| | 5 | "Conditional Random Fields" OR "CRF" AND "Function" AND "Analysis" AND "Citation" |
| | 6 | "Sentiment Analysis" AND "Citations" AND "Polarity Features" AND "Sentence Splitting" |
| | 7 | "scientific papers" AND "citation intent classification" AND "sentence extractions" OR "citation intent classification" |
| IEEE Xplore | 1 | ("Document Title": Citing Sentences) AND ("Document Title": Research Papers) OR ("Full Text Only": Citation Analysis) AND ("Document Title": Challenges) OR ("Document Title": Applications) AND ("Document Title": Sentiment Analysis) [Filters] year: 2010–2022 |

By applying the search queries, we obtained a total of 6801 articles. Due to the large volume of results, we decided to discard many papers. We applied the following approach: When a query returned more than 50 results, we saved the papers on the first results page; otherwise we saved all returned papers. We then discarded more papers, duplicates, and those that did not match the selection criteria we set. Table 2 shows the search results for each query in each database, as well as the articles we saved for further analysis. Most of the queries were performed on Google Scholar (11 queries). Table 3 shows the total number

of papers found per database, the total number of papers we saved, and the total number of papers we finally included in our review. Of the 6801 articles initially found, we saved 468 and finally included 37. A very large volume of papers was found via Google Scholar and ACL Anthology.

**Table 2.** Papers found and saved by search query.

| Digital Repositories/Databases | Number of Query | Papers Found | Papers Saved |
|---|---|---|---|
| Springer | 1 | 16 | 16 |
| | 2 | 43 | 43 |
| | 3 | 17 | 17 |
| | 4 | 15 | 15 |
| | 5 | 10 | 10 |
| Google Scholar | 1 | 53 | 10 |
| | 2 | 768 | 10 |
| | 3 | 651 | 10 |
| | 4 | 508 | 10 |
| | 5 | 305 | 10 |
| | 6 | 51 | 10 |
| | 7 | 17 | 17 |
| | 8 | 6 | 6 |
| | 9 | 62 | 10 |
| | 10 | 10 | 10 |
| | 11 | 246 | 10 |
| Semantic Scholar | 1 | 783 | 10 |
| | 2 | 62 | 10 |
| | 3 | 12 | 12 |
| Science Direct | 1 | 205 | 25 |
| | 2 | 353 | 25 |
| ACM | 1 | 36 | 36 |
| | 2 | 21 | 21 |
| MDPI | 1 | 30 | 30 |
| ACL Anthology | 1 | 6 | 6 |
| | 2 | 585 | 10 |
| | 3 | 782 | 10 |
| | 4 | 67 | 10 |
| | 5 | 702 | 10 |
| | 6 | 4 | 4 |
| | 7 | 51 | 10 |
| IEEE Xplore | 1 | 324 | 25 |

**Table 3.** Papers found, saved, and included in the review by Database/Digital Repository.

| Digital Repositories/Databases | Papers Found | Papers Saved | Papers Included |
|---|---|---|---|
| Springer | 101 | 101 | 8 |
| Google Scholar | 2677 | 113 | 10 |
| Semantic Scholar | 857 | 32 | 3 |
| ScienceDirect | 558 | 50 | 1 |
| ACM | 57 | 57 | 3 |
| MDPI | 30 | 30 | 2 |
| ACL Anthology | 2197 | 60 | 8 |
| IEEE Xplore | 324 | 25 | 2 |
| Total | **6801** | **468** | **37** |

In the process of systematically reviewing the existing literature, in addition to using reliable scientific databases, we also included papers discovered through citations of the included articles as well as work-projects from relevant websites. The selection of these

papers was shaped by their contribution to strengthening and deepening our review. This approach guarantees transparency in the methodology and source selection, ensuring that each incorporated paper or source contributes substantially to the understanding and interpretation of the research area of interest. At this point, we should mention that papers found via citations (as well as websites) are not considered in the PRISMA methodology, although we did include them in our review.

Figures 2 and 3 show the number of papers found, saved, and finally evaluated (bar graph), as well the percentages of papers included in the review (pie graph). Table 4 shows the number of all types of publications included in the review (Journal Article, Conference Paper, Website). Table 5 shows the publication types of only the papers found in the databases we used.

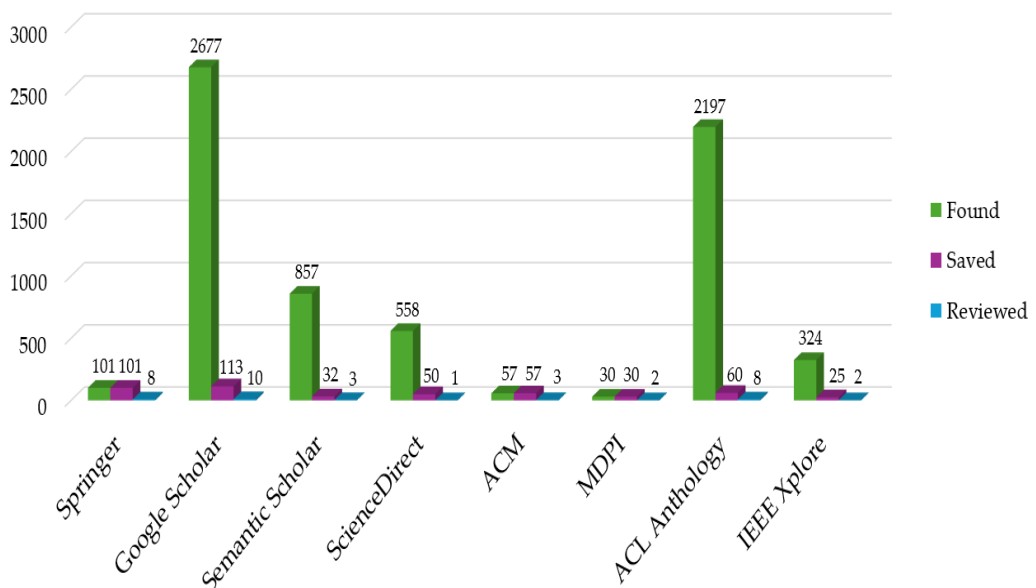

**Figure 2.** Articles found, saved, and included in the review.

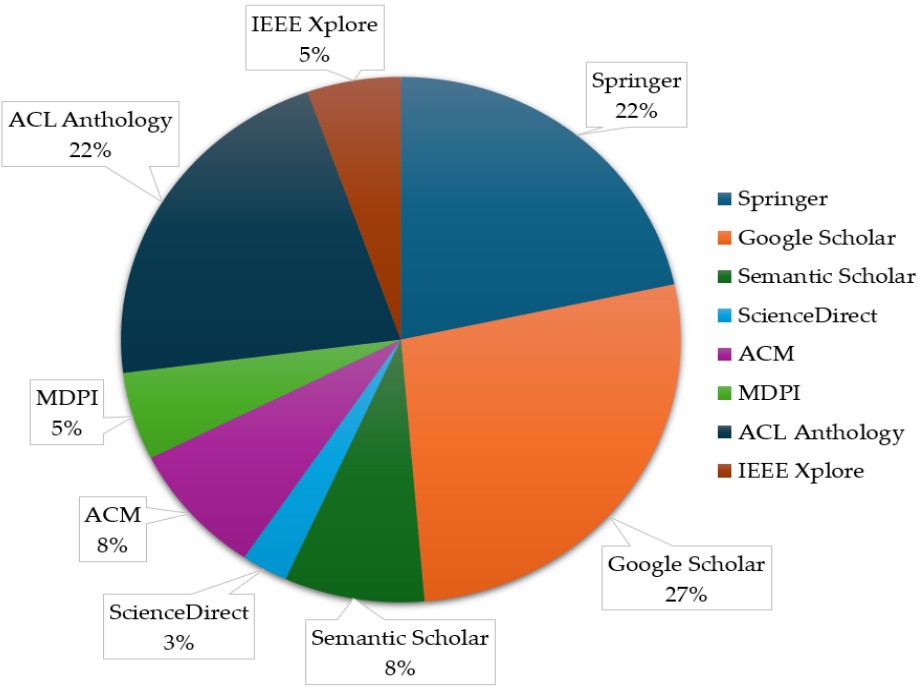

**Figure 3.** Percentage of papers included in our review by Digital Repository.

Figure 4 shows the percentages of publication types, with websites taking the smallest share. An equal number of papers are published in conferences and journals. Figure 5 shows the percentages of papers found in databases. Most of the papers are publications in conference proceedings.

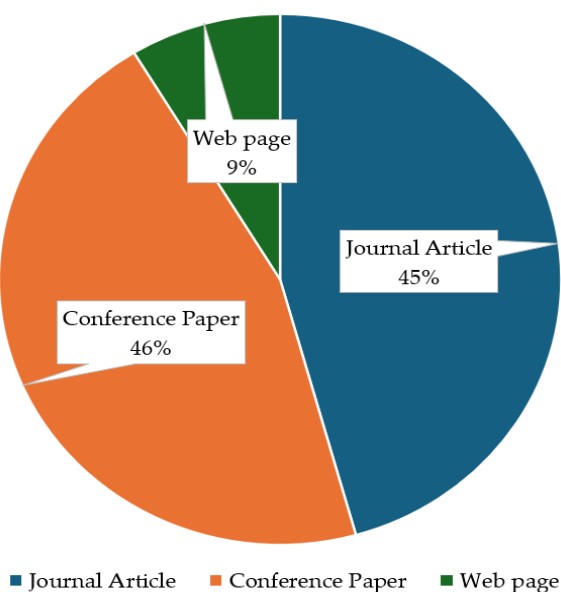

**Figure 4.** All types of publications of reviewed papers (includes papers found in papers we included in the review from the Digital Repositories).

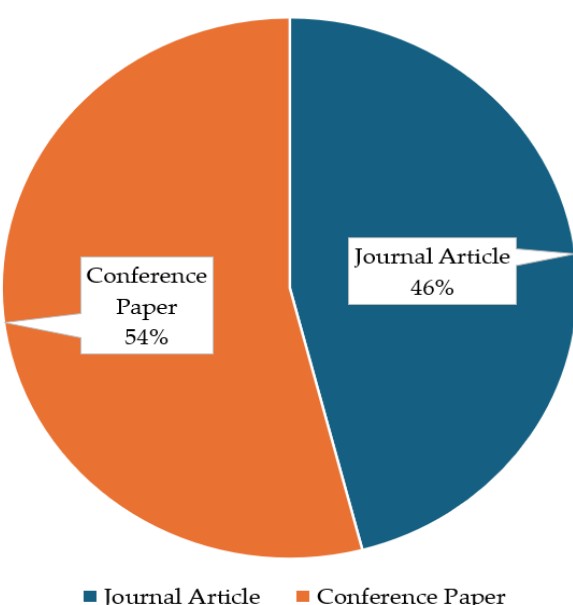

**Figure 5.** Publication types of reviewed papers (only in Digital Repositories).

**Table 4.** All types of publications included in the review.

| Publication Type | Number of Papers |
|---|---|
| Journal Article | 21 [1] |
| Conference Paper | 21 [1] |
| Website | 4 |
| Total | **46** |

[1] Four (4) Journal Articles and one (1) Conference Paper were found in the papers we have included in the review from the Digital Repositories.

**Table 5.** Publication types of only the papers found in the Databases/Digital Repositories.

| Publication Type | Number of Papers |
| --- | --- |
| Journal Article | 17 |
| Conference Paper | 20 |
| **Total** | **37** |

Figure 6 shows in detail the steps we followed according to the PRISMA Search Methodology. All steps were recorded, from the identification of papers found in digital libraries to the inclusion of the final papers in our review. At intermediate stages, we recorded the reasons for rejecting the papers.

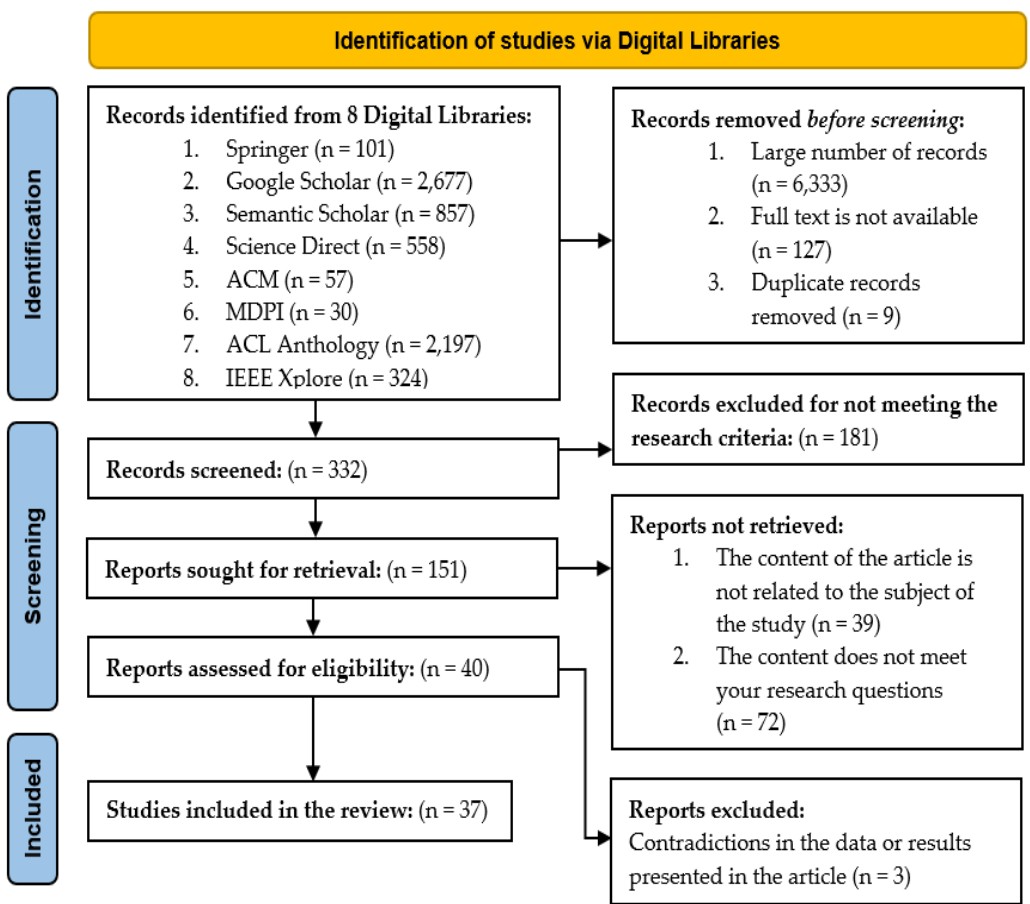

**Figure 6.** Papers retrieval steps (PRISMA Searching Methodology).

## 3. Literature Review

In this section, we focus on Sentiment Analysis and Scientometrics, presenting significant works conducted in these fields. Subsequently, we examine studies on scientific publication analysis related to classification and citation recommendation.

### 3.1. Sentiment Analysis

Before the introduction of Machine Learning and the so-called Transformer models in NLP, the process of detecting and understanding emotions in texts relied mainly on the use of specific dictionaries containing words with specific emotional values or tendencies. A prominent benchmark is the dictionary created by the researchers [9], which includes more than 2000 words categorized according to emotion polarization (positive, negative, or neutral emotion), objectivity, and Ekman's six basic emotions [10]. In addition, [9] used the Twitter API over a two-day period in March 2014, collecting 250,000 tweets written

in English and applying an ensemble Machine Learning algorithm that combines the predictions of several models to produce a more reliable prediction. In their experiments, this algorithm achieved an excellent average accuracy of 81.81%.

S. Symeonidis et al. [11] used the above dictionary to perform sentiment identification based on data from the social network Twitter in order to identify the sentiment emerging from the most popular topics (hashtags). The aim of this paper was to conduct an analysis of sentiment by covering Ekman's key emotions and not necessarily to identify polarity. They applied methodologies such as Arithmetic Mean, Quadratic Mean or Root Mean Square (RMS), Maximum, and CombMNZ. As statistical measures, they used the Pearson and Kendall correlation coefficients, where the highest Pearson score recorded was 0.26 for "Happiness" and the corresponding Kendall value was 0.22 for the same emotion.

Also, researchers P. Tsantilas et al. [12], utilized a different dictionary that consisted of at least 6000 words which are classified as positive or negative. In this case, the goal was reputation management, and a rule system was used to categorize sentiment in a dataset of more than 2000 texts; the accuracy of this methodology approached 64%. For polarity identification, they described an application for text analysis known as PaloPro, which combines several technologies, one of which is the OpinionBuster system, for extracting named entities. Finally, data were collected from a wide range of sources, including news from two Greek newspapers (Real News and Kathimerini), and posts on Facebook and Twitter.

More advanced methods use Machine Learning algorithms, while many different approaches can be found in the literature. The main divergences lie in the creation of so-called word embeddings, as well as in the choice of architecture and model parameters.

The resources [13,14] trace sentiment in a database of reviews in stores on the Skroutz platform, and they are an additional important source. In these sources, Neural Networks are used, where in [13], the researcher creates a Deep Neural Network by introducing an embedding layer, which transforms the multidimensional input into smaller dimensional vectors and achieves 92% accuracy. In [14], a version of the Bidirectional Encoder Representations from the Transformer (BERT) model is used, with 96% accuracy being achieved.

Similarly, the study [15] uses a Dataset for sentiment analysis of product reviews written in Greek, which includes less than 500 sentences classified as positive or negative, taken from the Skroutz website. This researcher uses two traditional Machine Learning algorithms: Support Vector Machine (SVM) and Naïve Bayes (NB). He combines SVM with Unigram features and the Term Frequency-Inverse Document Frequency (TF-IDF) technique. He also uses Unigram and Bigram features with NB by applying and deleting Stopwords. In addition, the researcher also considers a variant of the BERT model. With this small dataset, the researcher manages to achieve an excellent 97% accuracy with BERT over four training epochs. Regarding the SVM and NB models, in the case where all words were used as features, SVM scored 87% accuracy, followed by NB with 86%. When using Unigrams, SVM again prevailed with 86% accuracy, while NB achieved 84% accuracy. As for the Bigrams features, only the NB algorithm was used, featuring an accuracy of 89%. To improve the accuracy of NB, the paper tested its use with the help of the Stopwords deletion technique, where, in combination with Unigrams and Bigrams, they achieved 87% and 89% accuracy, respectively. Finally, another experiment was conducted in which SVM was used in combination with the technique of estimating the importance of a word in a text (TF-IDF). The result was satisfactory, as the accuracy reached 92%. From the experiments conducted in [15], a clear picture emerges of the dynamics that Transformer models, such as BERT, incorporate in regard to sentiment analysis.

The contribution of [16] to the research community is also important. In this paper, we consider another Machine Learning methodology using the SVM algorithm on datasets expressing people's opinions in different languages. More specifically, the researchers consider a hybrid approach for sentiment prediction in which they use the Word2Vec methodology to generate word embeddings in combination with the use of dictionaries. Finally, by applying different combinations, they achieve an accuracy of 83.60% on a set

of user ratings (Dataset MOBILE-PAR: includes 1976 ratings for training and 3329 for testing), a performance that significantly stands out from the unsupervised methods of other researchers, where, according to [16], they achieved an accuracy of 78.05%. Due to its great potential, the Word2Vec model has been used in many NLP research projects, offering remarkable results.

Cui et al. [17] conducted research on product reviews online and classified them as either positive or negative. They examined at least 100,000 product reviews collected from Froogle (an early name of Google's product search service; it was renamed Google Shopping in 2007) and trained Passive—Aggressive (PA) algorithms, which are variations of SVM models, and Language Modeling (LM) algorithms, which calculate the probability of a text appearing based on the n-gram occurrence frequency. The best accuracy achieved was reported using the PA Classifier with n-gram features for n = 6, where the overall F1-score approached 90%. The use of more complex features, such as higher-order n-grams, seems to confirm that the accuracy of sentiment classification in product reviews can be improved, providing more detailed and satisfactory content analysis.

The paper [18] presents and discusses the use of the Word2Vec model for sentiment classification in Twitter posts about US airlines. The models used in this research are Logistic Regression (LR), Gaussian Naïve Bayes (GNB), Bernoulli Naïve Bayes (BNB), and SVM. In addition, the CBOW and Skip-Gram methods, two key approaches to Word2Vec, were examined. Skip-Gram attempts to predict neighboring words given a central word, while CBOW attempts to predict a central word based on its neighboring words. The best accuracy obtained by CBOW is for the SVM classifier at 70%, while Skip-Gram achieves a higher accuracy of 72% when combined with SVM and LR.

The research paper [19] discusses sentiment analysis of hotel reviews in the Indonesian language retrieved from the Traveloka website using Selenium and Scrapy detection libraries. This research achieved an average accuracy of 85.96% on 2500 review texts using a combined approach featuring Word2Vec and the Long Short-Term Memory (LSTM) model. More specifically, Word2Vec was used to generate the word embeddings from the hotel reviews, and these embeddings were then fed into the input of the LSTM model to classify them with a positive or negative polarity. The LSTM architecture has the advantage of being able to maintain an internal state (cell state) which acts as a memory that allows information to be stored for long periods of time while having the ability to forget information that is not useful.

Another very important contribution to the research community is the work of [20]. In this study, the researchers use a dataset that includes at least 7900 negative comments, more than 7000 positive comments, and over 44,000 neutral comments of varying length, all originating from different social media platforms. They perform tests on binary (2 classes: negative and positive) and three-class (3 classes: negative, positive, and neutral) classification, using Transformers models and other advanced architectures. They are particularly interested in three-class classification, with which they train a GreekBERT model, a PaloBERT model based on the Robustly Optimized BERT Pretraining Approach (RoBERTa), and a GreekSocialBERT model, which is an extension of GreekBERT. Although the dataset does not have balanced class-clusters, the researchers achieve an excellent performance, scoring 99% accuracy while using a Generative Pre-trained Transformer (GPT) model for binary classification. On the other hand, in the three-class case, the GreekSocialBERT model shows the highest performance, achieving 80% accuracy.

### 3.2. Scientometrics

An important branch of research dealing with the measurement, analysis, and evaluation of scientific activity is Scientometrics [21], which is often considered the science of science. The main difference between Scientometrics and Sentiment Analysis is that it uses mainly quantitative methods. The goal of Scientometrics is to evaluate the development of a field and the influence of scientific publications. It essentially monitors research, evaluating the scientific contribution of author-researchers, journals, and specific papers, as

well as evaluating the development and dissemination of scientific knowledge [22]. The researchers González-Alcaide et al. [23] used scientometric methods to identify the main research interests and directions regarding cardiomyopathy in the MEDLINE Database, one of the most well-known and authoritative databases in the field of medicine and health, which is under the auspices of the National Library of Medicine (NLM) of the United States. They identify research patterns and trends in Chagas' cardiomyopathy. Similarly, Mosallaie et al. [24] used scientometrics approaches to identify trends in cancer research, while Wahid et al. [25] applied scientometric methods and a comparative analysis to a group of authors to determine their scientific productivity. Additionally, [26] presented an alternative approach by mainly applying Convolutional Neural Networks (CNNs) to classify scientific literature. The model they proposed performed better compared to classical Machine Learning methods in terms of accuracy.

### 3.3. Scientific Citation Analysis (SCA)

### 3.3.1. Citation Contribution

In the world of scientific research, no research work is exclusively independent, as it is necessarily embedded in the literature of the respective research field. Citation-referencing, a vital element of this embedded structure, reveals the relationships and interactions between research articles, confirming the interactivity and ongoing debate within the scientific community. Beyond being just a reference method, citations have a critical role in the scientific literature, contributing to the ranking of various aspects, such as the ranking of research institutions and authors [5]. Citation analysis is at the core of bibliometrics, functioning as the science that studies these complex relationships between research articles. This systematic process through which authors cite the works of others creates a dense network of citations that is essential for the maintenance and advancement of scientific knowledge [5,27].

As mentioned above, sentiment analysis identifies and classifies opinions expressed in documents. Sentiment analysis of citations has attracted particular attention for two main reasons: First, to improve bibliometric metrics by focusing primarily on the quality rather than quantity of citations, with the aim of reducing bias and providing evidence-based support for writing. Second, to detect non-reproducible research, i.e., the identification of research papers or results that cannot be replicated or verified by other researchers, especially in the biomedical field, where unfavorable attitudes may be early indicators of the non-reproducibility of research, thus saving time and resources [28]. Therefore, although positive polarity citations have a significant impact on science, as they can enhance the validity and reliability of findings and even promote the reputation and career of researchers, the study by Catalini et al. [29], however, equally highlights that negative citations can also play an important role in science. Indeed, in some cases these citations can help to improve initial findings and aid in the development of a field, indicating the multidimensional importance of emotion analysis in scientific research. Often, however, due to their nature, such citations may simply not attract attention, and the information they offer may take some time to become widely known [29]. Therefore, observing the trajectory of negative citations, as well as the various motivations that lead to the citation of prior literature, is a very important process [29].

### 3.3.2. Text and Citation Preprocessing

Text Preprocessing before classification is a critical step in the process of extracting useful information and knowledge from the data. This process usually involves techniques such as tokenization, whereby a text is broken down into tokens; cleaning the text of unwanted elements, such as punctuation and other special characters; removing words without significant meaning (Stopwords); and converting to lower case. In addition, there are other important techniques, such as Lemmatization, where words in various forms are converted to their basic form (known as lemma), and Entity Recognition, where a system attempts to identify and categorize the names of people, organizations, places, etc.

within the text. Entity Recognition or Named Entity Recognition (NER) is particularly useful for structured organization of information, as it helps to further analyze the data and therefore is also part of the preliminary steps that prepare the text for more specialized Machine Learning techniques [5,18]. In addition, the Term Frequency-Inverse Document Frequency (TF-IDF) and Word2Vec techniques are also part of the broader text processing process. They refer to the phase of representing words in the form of vectors, which usually follows basic pre-processing. TF-IDF is a statistical method used in NLP to evaluate how important a word is in a document. The more often a word appears in a document, the greater its importance. Word2Vec is also a technique that generates word vectors using Deep Neural Networks. These vectors represent words in a continuous-dimensional space where words with similar semantic properties are close to each other. This allows the models to understand words based on their context of use and their relationship with other words [5,18].

At this point, it should be emphasized that there are significant differences in preprocessing plain text compared to a scientific text. These differences stem from the nature of the vocabulary, the structure of the text, and the complexity of the information. Scientific texts include technical terminology, so preprocessing must manage these concepts appropriately and preserve relevant terms rather than removing them as noise. Scientific texts include citations to other works that need to be recognized and managed differently from the plain text. To effectively preprocess scientific texts, there are several steps that can help to better manage and analyze the data. In terms of special character management, characters that are important in scientific terminology, such as mathematical formulas, should be preserved. In addition, in terms of identifying citation contexts, keywords should be kept that identify semantic citations such as expressions of the form "according to <author>" or the form "author et al.". This depends on the citation style used. Finally, in scientific texts, an excellent preprocessing technique is the NER procedure mentioned above. NER can improve the way words are represented within a document, identify entities, and extract information from a large volume of scientific articles. Finally, NER systems are very often combined with ontologies to identify categories of entities, moving beyond general labels such as "person" to more specific and scientifically relevant labels.

### 3.3.3. Citation Context Retrieval Methods and Classification

Retrieving citation context from scientific articles aims to understand and analyze the content. The process starts with the identification and extraction of sentences containing citations. This is usually achieved using NLP models and Machine Learning algorithms, such as SVM or Conditional Random Fields (CRFs), which analyze the text and identify areas that may contain citations. Once these areas are identified, the next phase is to interpret the content to apply further analysis techniques depending on the research objectives, such as the polarity assessment discussed in the previous sections. Figure 7 shows the basic steps of text classification.

Many researchers have used open-source tools or other techniques to retrieve and analyze citations. Awais Athar [30], in his research in 2011, studied Supervised Learning by applying the SVM model with n-grams, length 1–3, and other features to analyze citations. He chose the ACL Anthology Network [30–32] for data collection and analyzed a total of 8736 citation frames from 310 scientific articles via manual labelling methods, classifying each sentence into a category: positive, negative, or neutral. In addition, he separated the data into 1472 samples for training and 7264 samples for control, of which 6277 were classified as neutral, 743 as positive, and 244 as negative. Thus, an unbalanced data set was formed. The results of his experiments showed that applying such an approach is useful for identifying only explicit citations [2]. As evaluation metrics he reported macro-F1 and micro-F1 using 10-fold Cross Validation. The best results obtained were 76.40% and 89.80% for macro-F1 and micro-F1, respectively [30].

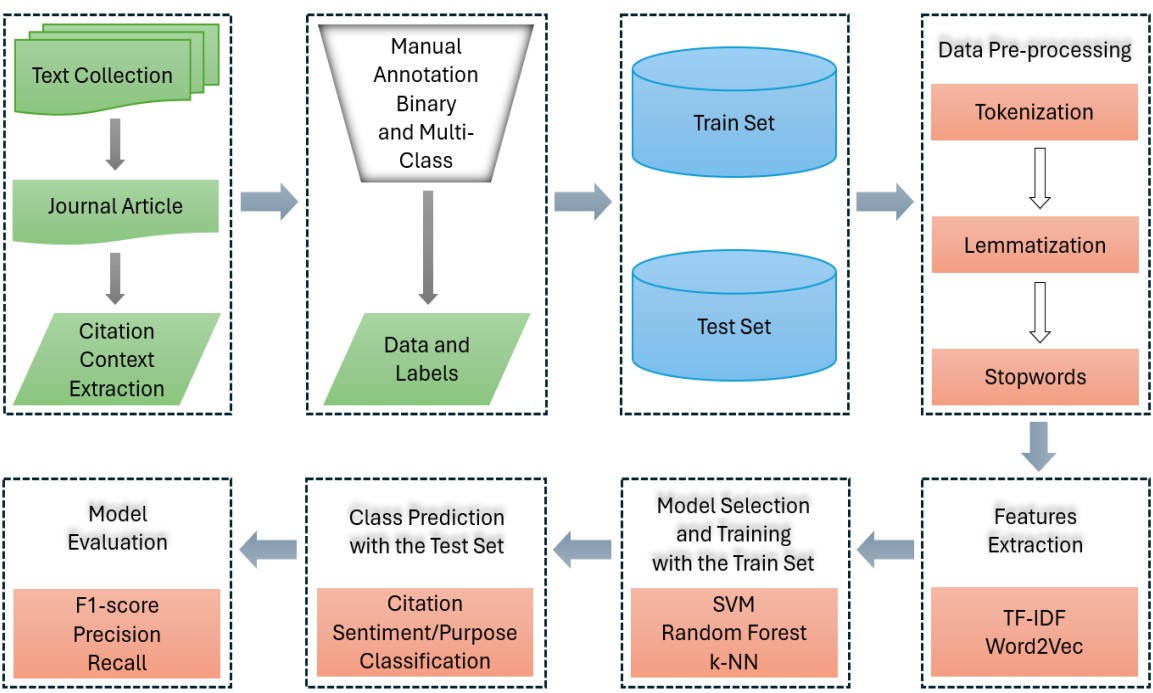

**Figure 7.** The basics steps for Text Classification with a focus on Citation Contexts.

Councill et al. [33], in their 2008 research, describe ParsCit, an open-source software tool for retrieving the citation context from research papers and analyzing literature strings. To enable comparison with other related tools, the researchers focused on literature analysis, meaning that they examined the references typically listed in the last section of a scientific article, ignoring the contextual contexts within the text. At the core of ParsCit is a pre-trained CRF model that is used to label the tokens of strings. Furthermore, it offers additional functionality using state-of-the-art Machine Learning models and heuristics to achieve high accuracy in text segmentation, as well as in string recognition and retrieval. Also, the software comes with utilities to run as a standalone or as a Web service. One of the key works of [33] was the comparison of ParsCit with an older CRF-based system proposed by Peng and McCallum in 2004 [34]. This system was the source of the research of [33]. The dataset they used was Cora [35], which is one of the earliest works in text analysis. This dataset created a template with 200 reference samples collected from a variety of scientific publications in the field of computer science [33]. Each of these references was divided into thirteen distinct categories: "author", "book title", "date", "publisher", "organization", "journal", "location", "notes", "pages", "publisher", "technology", "project title", and "volume" [33]. The results showed the superiority of ParsCit (with Average F1-score: 95%) over Peng CRF (with Average F1-score: 91%) [33].

Due to the effectiveness and widespread use of the SVM algorithm, the team of Ezra et al. [36] successfully applied this algorithm to classify citation sentences within the text. According to them, existing bibliometric measures usually provide quantitative indicators of how good a scientific paper is. However, this does not necessarily mean that they reflect the level of quality of the work exposed in the research. For example, when calculating a researcher's h-index, every incoming citation is considered in the same way, ignoring the possibility that some of them might be negative [36]. Thus, researchers [36] proposed the use of NLP techniques to add a qualitative aspect to bibliometrics. Specifically, they analyzed the citation contexts of scientific articles obtained from the ACL Anthology Network [32] and applied supervised Machine Learning methods to determine the purpose and polarity of the citations. To categorize purpose, they used six category-classes: "Critique", "Comparison", "Use", "Documentation", "Base", and "Other". For their experiments, they applied several classification models, including LR, Naïve Bayes Classifier, and SVM. The

researchers do not present the results of all the algorithms; however, they do highlight SVM, which achieved the highest Accuracy of 70.50%, while macro-F1 reached 58%. For the citation polarity classification, only the results for the SVM model are also presented. Two experiments were conducted: in the first one, only explicit citations were used without considering any other context of the text, while in contrast, the second experiment used the wider context surrounding a sentence. This broader context does not exclusively involve implicit citations; it simply includes those sentences that are close to the referencing sentence and are considered important or relevant by human evaluators for understanding the meaning of the citation. The results noted with SVM are Accuracy 74.20% and 84.20% and macro-F1 62.10% and 74.20% for the first and second experiments, respectively [36]. The findings of the study point out that incorporating the wider context of the citation significantly contributes to improving classification accuracy (especially in the categories with subjective nature, and particularly in the negative category). This can be seen through the improvement in the Recall metric of the negative category, which, while only reaching 71.10% in the first experiment, improves by 10 points to approach 81.10% in the second experiment [36].

A different approach from the above papers that focuses on literature analysis and the purpose/polarity classification of citations is the research by Kumar et al. [37], who applied Supervised Learning using Maximum Entropy (ME) and SVM classifiers. Their goal was to determine whether a sentence in an article is a citation to another article or not, thereby making it a Binary Classification problem. They used the ACL Anthology Reference Corpus (ACL ARC) [32,38] for their experiments. The ACL ARC numbered about 10,921 articles by February 2007, and the researchers were able to retrieve features from a total of 955,755 sentences. Then, for citation identification, they identified 112,533 sentences as instances containing citations (positive samples), followed by subsequent processing to remove citation markers (e.g., IEEE styles such as [1,2] or APA such as Schmidt, 2017) from them. The remaining 843,222 sentences were classified as sentences that did not constitute citations (negative samples) [37]. Thus, they formed a dataset and applied the 10-fold Cross Validation evaluation method by separating the data into two parts: 90% for training and 10% for testing. This procedure was repeated 10 times to obtain the results of the evaluation [37]. According to their results, the lowest accuracy was noted in the "Bigram" feature for both models. ME achieved an accuracy of 82.70%, while SVM reached 85.10%. On the other hand, the highest accuracy was achieved on the features "Proper Noun" and "Previous and Next Sentence". Both ME and SVM achieved the same maximum accuracy of 88.20% in both the above features [37]. Also noteworthy is the conclusion drawn about the size of the training data. By changing the volume of this data, variations in the performance of the models can be discerned; however, the ME shows more variation mainly in the accuracy of the "Unigram", "Bigram" and "All" features. This means that the accuracy of these features depends on the volume of training data, and it follows that, the larger the size of these data, the higher the classification accuracy that can be achieved [37].

The features retrieved by [37] to construct their classifiers are presented in more detail below:

- *Unigram*. Unigram refers to a model of language analysis where the key element is the individual word. In this framework, each word in a sentence is considered an independent element or feature. In NLP, Unigrams are used to analyze and understand texts based on the individual words that make up the texts [37].
- *Bigram*. Bigram is a linguistic unit consisting of two consecutive words. In NLP, Bigrams are used to understand the relationships and structures created between two consecutive words in a sentence. This helps in analyzing the language flow and word combinations that are frequent in each text [37].
- *Proper Nouns*. These are nouns that describe the names of people, places, and organisms. These features are of great importance in the detection of referential sentences, as it is known that such sentences tend to focus on different institutions, specific scientists, and the systems they have developed [37].

- *Previous and Next Sentence*. This is information about neighboring sentences. For example, if a sentence follows a sentence with a citation, it may continue the discussion of the same topic, so it is less likely to include an additional citation [37].
- *Position*. The position attribute provides information about the part of the document in which a sentence appears. These attributes are important, as sentences appearing in certain sections have different probabilities of containing a citation. For example, sentences in the middle or at the end of a research article are more likely to discuss authors' works, evaluations, or experiment results, so they are considered less likely to be areas with citation compared to the beginning of the article, where authors often discuss and acknowledge previous work [37].
- *Orthographic*. This group of features looks at various morphological elements in sentences, including the specific orthographic forms used. Sentences that include numbers or single capital letters tend to be more suggestive of citation sentences, as they may indicate comparative figures or the initial letters of the name of the authors of the papers being referenced [37].
- *All*. Includes all the above features.

One of the main difficulties in Machine Learning approaches is their dependence on the correct choice of features [2], at least as far as Sentiment Analysis in texts and scientific citations are concerned. Therefore, feature extraction methods are not effective in some cases, as is the case with recognizing the negation or opposite meaning of a sentence. For example, the sentence "I hate violence" might not elicit any negative emotion; however, a Machine Learning model might, due to the presence of two negative words, classify it as a sentence with a negative polarity. These are the limitations that Deep Learning models are called upon to address, as they can produce semantic representations. According to [2], not much research has been conducted regarding the Sentiment Analysis of scientific citations with Deep Learning models; however, they propose the implementation of Recurrent Neural Networks (RNNs) to test the effectiveness, as they show good performances in regard to interpreting semantic content.

One research that examines Neural Networks for Sentiment Analysis in citations is the work of Munkhdalai et al. [39]. Their study describes the development of a new Neural Network model called Compositional Attention Network (CAN). They use data from PubMed Central, focusing on function categorization and sentiment analysis in four classes: "Negational", "Confirmative", "Neutral", and "Do Not know". Specifically, they selected 5000 citation sentences from 2500 random articles, then organized a tagging scheme for these sentences where each sentence was tagged by five human annotators. Finally, they constructed two datasets for training and evaluation. The first dataset consisted of labels on which at least three of the five annotators agreed (Three Label Matching). This resulted in 3624 citations for sentiment analysis. To construct the second dataset, most of the opinions of the five commentators were relied upon. In other words, a label was chosen for each citation text only if that label was decided by a majority of the five commentators (Majority Voting). This means that, even if only two commentators agreed on a label, it would be entered into the dataset because it represented a clear majority, as the other three labels differed. As a result, a total of 4423 citation suggestions for sentiment classification were entered into the second dataset. It becomes obvious from the above that the Majority Voting approach is more lenient compared to the Three Label Matching method. In addition, the researchers applied models such as LSTM, Bi-LSTM, and attention models. CAN shows significant improvement in accuracy, especially when additional sentence context information is included. For sentiment analysis, the LSTM model combined with CAN achieves the highest accuracy compared to the other models (76.04% for Majority Voting and 78.10% for Three Label Matching), showing its superiority in handling more information and providing better representations of the data. It should also be mentioned that the study of [39] also used the SVM model, which showed low generalization to new data as it scored the lowest accuracy in both methods (75% for Majority Voting and 71.95% for Three Label

Matching) compared to the Neural Network models, which highlights the superiority of Deep Learning.

Progress in the field of Deep Learning led to the creation of the Transformer language models. These models are a powerful class and have proven their effectiveness in many AI applications. These models were originally introduced to solve problems in the NLP domain, such as text generation and entity recognition. The main feature of Transformers is their ability to consider the semantic dependencies between words in a text without the use of traditional recursive architectures. This is achieved through mechanisms that focus on parallel processing of information in large sequences of data, such as the Attention Mechanism. Important research on sentiment recognition using Transformer models was conducted by researchers Dahai Yu and Bolin Hua [40]. In their study, they emphasized the importance of pre-trained models such as BERT, which was trained on general texts from the internet, and SCIBERT, which is a variant of BERT and was trained using scientific articles. According to [40], SCIBERT is considered more suitable for applications related to the scientific and academic community, such as the classification of scientific texts and the recognition of emotions in them. After a detailed investigation, it was found that several sentiment analysis studies did not disclose the datasets, while, in other cases, the available datasets proved to be of low quality [40]. To further improve the accuracy of content-level training, the researchers decided to use the SCICite dataset proposed by Arman and colleagues [41]. This dataset included a training set of about 10,000 citations and a control set of about 1000 citations, which were classified into three categories in terms of intent: "Method", "Background" and "Result" [40]. They also considered the dataset proposed by Athar in [30] and, after extracting about 1000 citations from SCICite, they enhanced Athar's dataset. Finally, the aggregated dataset consisted of 7912 suggestions, including 1237 positive, 347 negative, and 6328 neutral [40]. To perform their experiments, in addition to the two pre-trained models (BERT, SCIBERT) used as a basis, they designed and proposed the DictSentiBERT model, which adapts the Dictionary-based Attention Mechanism and applies emotion categorization of scientific citations [40]. In addition, four other models, LSTM, FeedForward NN (FNN), TextCNN, and Self-Attention, were tested. The models were trained on an RTX A4000 processor with 16 GB of memory and a maximum number of epochs of 50. During an epoch, the data was split into an 80% for the training set and a 20% for the test set. The Batch Size and Learning Rate parameters were set to 32 and $5 \times 10^{-6}$, respectively. AdamW was used as the optimizer, and cross-entropy was used as the loss function [40]. From the data presented in Table 6, the FNN, LSTM, TextCNN, Self-Attention, and DictSentiBERT models based on both BERT and SCIBERT showed high Accuracy, with DictSentiBERT achieving the highest accuracy (BERT 93.49% and SCIBERT 95.20%). Additionally, the BERT model showed an average accuracy of 91.23% and an average macro-F1 value of 74.60%. In contrast, SCIBERT showed even better results, with an average accuracy of 94.80% and an average macro-F1 value of 85.20%. This finding suggests that SCIBERT, which, as mentioned, was specifically trained on scientific texts, is more suitable for analyzing and categorizing emotions in citation texts. Furthermore, the improved performance of DictSentiBERT indicates the advantage of incorporating a sentiment lexicon into the model [40].

**Table 6.** Experimental results. Accuracy and macro-F1 (%) [40].

| Models | BERT | | SCIBERT | |
| --- | --- | --- | --- | --- |
| | Accuracy | Macro-F1 | Accuracy | Macro-F1 |
| FNN | 93.05 | 80 | 95.14 | 86 |
| LSTM | 93.11 | 80 | 94.63 | 84 |
| TextCNN | 83.20 | 52 | 94.57 | 86 |
| Self-ATTENTION | 93.30 | 80 | 94.44 | 84 |
| DictSentiBERT | **93.49** [1] | 81 | **95.20** [1] | 86 |

[1] Max Accuracy for DictSentiBERT.

The dynamics of Transformer models were also highlighted in the study by Ning Yang et al. [42], which analysed the effectiveness of BERT-based methods for identifying scientific data citations while focusing on information extraction from bioinformatics texts and citation recognition as a Binary Classification problem. The texts were obtained from PubMed Central (PMC), where 35 journals were collected as data sources and 38,931 full-text documents were retrieved. The paper classified the diverse forms of text citations into the categories of "scientific data citations" and "non-scientific data citations"; these two categories were treated as positive and negative, respectively (Binary Classification). In the end, 3067 citations (positive samples) and 12,869 citations (negative samples) were obtained. The study compared the performance of some models, such as SCIBERT discussed above, with classical Machine and Deep Learning models. The study also found that BERT-based models, especially BioBERT, perform better compared to other models. For their experiments, in addition to SCIBERT, BERT and BioBERT, classical models such as, Decision Tree model, Random Forest model, TextCNN, and TextRCNN were used. In Table 7, we present the results, which show the superiority of the BERT based models. Precision, Recall, and F1-score metrics are also shown. Of significant interest is the BioBERT model proposed by Lee et al. [43], which is based on BERT and applied to the biomedical domain (which is closely related to the field of bioinformatics). This makes it a high-performance model which, in the study of [42], scores the highest Recall.

**Table 7.** Models and Metrics. Precision (%), Recall (%), F1-score (%) [42].

| | METRICS | | |
|---|---|---|---|
| **Models** | **Precision** | **Recall** | **F1-Score** |
| Random Forest | 82.80 | 71.60 | 75.20 |
| Decision Tree | 75 | 75.40 | 75.20 |
| TextCNN | 86.40 | 75.60 | 79.40 |
| TextRCNN | 84.20 | 76.50 | 79.50 |
| BERT | 86.90 | 82.70 | 84.60 |
| SCIBERT | 86.70 | 84.10 | 85.30 |
| BioBERT | 85.70 | **84.90** [1] | 85.30 |

[1] Max Recall for BioBERT.

Finally, this research [42] demonstrates that Machine and Deep Learning techniques are successful in detecting and classification scientific citations. Moreover, the findings of this study support that Deep Learning outperforms traditional models by achieving higher generalization and performances, as it considers the semantic features of a document. The capability of these models makes them an important tool in natural language analysis and processing, offering significant potential for accurate interpretations of information.

### 3.3.4. Citation Recommendation

The development of a Citation Recommendation System (CRS) can help researchers discover additional research relevant to their topic. Through sophisticated algorithms and Machine Learning models, such a system can recommend citations that are closely related to the content of the article. By highlighting the most relevant citations, researchers can enhance the validity and relevance of their work. When writing research articles, there are often instances where previous research needs to be referenced, but there is no certainty in selecting cited sources. In their study, He et al. [44] propose a context-aware CRS. Creating high-quality citation proposals can be significantly challenging as the citations proposed must be relevant to the topic of the article and adapted to the specific contexts where they are used. The main idea of [44] is therefore to design a new non-parametric probabilistic model that can evaluate the relevance of a citation context and a paper. Similarly, the issue of citation recommendation was also addressed by Silvescu et al. [45]. In their research,

they examined the challenges of discovering relevant citations by focusing on the use of the Singular Value Decomposition (SVD) technique compared to Collaborative Filtering (CF) methods. The results of their experiments showed the superiority of the proposed SVD approach, which achieved significant success compared to CF methods. Their paper also discussed the creation of a new dataset from the CiteSeer Digital Library [46] for experimentation and evaluation on more advanced recommendation models.

The above research highlights the importance of the evolution in citation recommendation technology, offering more interesting, comprehensive, and relevant information to researchers.

## 4. Challenges in Sentiment Analysis

Sentiment Analysis in text, in general, faces several challenges that range mainly from technical issues to semantic aspects. Some of the most basic challenges are discussed below:

- *Syntax errors*. Natural language is complex, and people often make syntactic errors which can make it difficult to process language automatically.
- *Multiple meaning*. Words can have multiple meanings depending on the context in which they are used, which can create confusion and misinterpretation. The use of complex vocabulary usually makes it difficult to understand the information. For example, in a text containing the phrase "It was terribly good", the word "terrible" usually has a negative connotation; however, in this phrase it is used to reinforce a positive adjective, "good", which can confuse automated sentiment analysis systems.
- *Variety and style*. Texts in general can include various types of written expression, such as literature, essay, narrative, journalism, and many others, each with its own style and mode of expression.
- *Complexity*. Natural language in general is complex and multidimensional, with sarcasm, allegory, hyperbole, and other elements adding considerable complexity to the analysis of emotions [47]. Irony and innuendo often escape analysis by automatic systems, which can lead to misunderstandings and misinterpretations of emotional tones in research.
- *Subjectivity*. As the understanding of emotions is subjective, different people may interpret the same texts differently [47].
- *Ambiguity*. Dealing with vague or contradictory statements in texts is a very important challenge.
- *Cultural differences*. Cultural and dialectal differences can affect the way emotions are expressed, making analysis difficult for systems not trained in different languages or cultures [47]. For example, in some cultures, the expression of anger may be less direct or intense compared to others. This may affect the accuracy of emotion analysis models that have not been trained to recognize such variations.
- *Spam detection*. The content present in messages can be complex, which makes it difficult to identify as spam. Moreover, the amount of data to be analyzed is huge, making spam detection resource intensive [47].
- *Language evolution*. Natural language is dynamic and constantly evolving, requiring a corresponding evolution of methods and systems for emotion analysis.

There are significant differences in the challenges encountered when analyzing emotions in texts compared to those encountered in scientific publications. When examining scientific citations, emotion identification is a complex challenge due to the specialized nature of language, the need to accurately understand emotional nuances, and the complexity of scientific concepts. This requires the development of advanced algorithms that can adapt to the constant changes in the field of linguistic and scientific development. Some of the fundamental challenges are discussed below:

- *Complexity and complex vocabulary*. Scientific citations often include specialized vocabulary and technical terms that may not express emotions in the traditional way.
- *Abstraction*. The use of language is often more abstract and less direct, resulting in a lack of strong feelings towards the reported research [2].

- *Multilingualism*. Citations can be written in multiple languages, increasing the complexity of sentiment analysis due to differences in grammar, syntax, and affective expressions that are specific to each language [2].
- *Context and social environment*. Understanding the context and social environment in which a scientific article was written is essential for accurate analysis of emotions.
- *NLP methods*. The development of algorithms that can recognize and interpret polarity in scientific texts requires advanced NLP techniques.
- *Lack of datasets*. There are not many datasets available that are labeled either for purpose or for citation polarity [2]. The creation of a database that is enriched with citation contexts to serve later in the training of a model capable of recognizing citations in scientific texts (while, at the same time, distinguishing their polarity) emerges as a significant challenge.
- *Stop words*. As mentioned, these are a category of words that are usually removed from the data in NLP applications. These words often include prepositions, links, and other common words that do not add significant meaning to the essence of a document. However, in scientific texts, the absence of some of these words can negatively affect classification performance [2].
- *Exporting a citation context*. Identifying the right context is an important issue. The contexts derived are varied. Some researchers focus on extracting a single sentence, while others extract entire paragraphs. This diversity makes accurate extraction an important and complex process [2].
- *Citation label*. How a class is assigned to a citation sentence is of great importance. In many cases this process is undertaken manually, making it difficult to label large datasets. Therefore, the process of automatic tagging in such texts is a very important challenge [2].
- *Words of denial*. The role of negation words is crucial in determining the emotional direction of a citation context. Identifying and handling negation is a difficult process and continues to be a significant challenge, as it can result in reverse polarity [2].

Below, we present a concise table that compiles and examines the primary challenges encountered, the Machine Learning models, the management of available resources and datasets, as well as the performance analysis through the experiments of the studies investigated (Table 8). Papers that do not provide enough information, such as models, datasets, and experimental results, were not included in the table.

**Table 8.** Comprehensive Overview of Machine Learning Challenges, Data Management, and Performance Insights.

| Authors, Year | Challenges | Models, Techniques | Datasets, Data Sources | Experimental Results |
|---|---|---|---|---|
| H. Cui et al., 2006 [17] | Sentiment Analysis in Product Reviews | Passive-Aggressive (PA) Language Modeling (LM) | Froogle | Accuracy: 90% |
| I. G. Councill et al., 2008 [33] | References Extraction, ParsCit vs. Peng CRF Comparison | ParsCit, Peng | CORA Dataset | ParsCit micro-F1: 95% Peng CRF macro-F1: 91% |
| K. Sugiyama et al., 2010 [37] | Citation Recognition, Binary Classification | Max Entropy (ME), Support Vector Machine (SVM) | ACL Anthology | Min Accuracy (Bigram Feature) ME: 82.70% SVM: 85.10%, Max Accuracy (Proper Noun and Previous and Next Sentence) ME and SVM: 88.20% |

**Table 8.** *Cont.*

| Authors, Year | Challenges | Models, Techniques | Datasets, Data Sources | Experimental Results |
|---|---|---|---|---|
| A. Athar, 2011 [30] | Polarity Analysis in Explicit Citations | SVM, WEKA | ACL Anthology and Resources [1] | macro-F1: 76.40% micro-F1: 89.80% |
| A. A. Jbara et al., 2013 [36] | Citation Context Analysis, Citation Purpose Classification, Citation Polarity Classification | SVM, Logistic Regression (LR), Naïve Bayes (NB) | ACL Anthology | SVM only Purpose Class. Accuracy: 70.50% macro-F1: 58%, Polarity Class. Explicit Accuracy: 74.20% macro-F1: 62.10%, Polarity Class. Wide Content Accuracy: 84.20% macro-F1: 74.20% |
| A. Tsakalidis et al., 2014 [9] | Tweets Extraction, Polarity Analysis, Feature Extraction | TBR, FBR, LBR, CR, Twitter API, Ensemble Algorithm | Resources [2] | Accuracy: 81.81% |
| P. Tsantilas et al., 2014 [12] | Sentiment Analysis, Named Entity Recognition | PaloPro [3] OpinionBuster [4] | Real News [5] Kathimerini [6] Facebook, Twitter | Accuracy: 64% |
| S. Symeonidis et al., 2015 [11] | Greek tweets Extraction, Sentiment Analysis | Maximum, CombMNZ, Arithmetic Mean, Quadratic Mean, Twitter Streaming API | Dataset with Greek tweets | Pearson Correlation 0.26 Kendall Correlation 0.22 |
| T. Munkhdalai et al., 2016 [39] | Citation Function Classification, Citation Sentiment Classification | Compositional Attention Network (CAN) | PubMed Central (PMC) and Resources [7,8] | Citation Function F1-score Bi-LSTMs + CAN Majority Voting: 60.67% and Three Label Matching: 75.57%, Citation Sentiment F1-score LSTM + CAN Maj. Vot.: 76.04% and T. L. Matching: 78.10% |
| M. Giatsoglou et al., 2017 [16] | Sentiment Analysis | Word2Vec, Lexicon Based | Mobile—PAR | Accuracy: 83.60% |
| J. Acosta et al., 2017 [18] | Sentiment Analysis of Twitter Messages | LR, Gaussian Naïve Bayes (GNB), Bernoulli Naïve Bayes (BNB), SVM, CBOW, Skip-Gram, Word2Vec | Twitter, Kaggle [9] | Accuracy CBOW + SVM: 70% Skip-Gram + SVM: 72% Skip-Gram + LR: 72% |
| P. Muhammad et al., 2021 [19] | Sentiment Analysis | Word2Vec, LSTM, Selenium, Scrapy | Traveloka Travel Platform [10] | Accuracy: 85.96% |
| G. Alexandridis et al., 2021 [20] | Polarity Analysis in Greek Social Media | Transformers, GreekBERT, PaloBERT, RoBERTa, GreekSocialBERT, GPT | Greek Social Media | Binary Classification GPT Accuracy: 99% Multi Classification GreekSocialBERT Accuracy: 80% |
| N. Avgeros, 2022 [13] | Sentiment Analysis | Neural Networks | Database from Skroutz | Accuracy: 92% |
| N. Fragkis, 2022 [14] | Sentiment Analysis | BERT Model | Database from Skroutz | Accuracy: 96% |

**Table 8.** *Cont.*

| Authors, Year | Challenges | Models, Techniques | Datasets, Data Sources | Experimental Results |
|---|---|---|---|---|
| D. Bilianos, 2022 [15] | Sentiment Analysis | SVM, NB, TF-IDF, BERT | Resources [11] | NB + Bigrams + Stopwords Accuracy: 89%, SVM + TF-IDF Accuracy: 92%, BERT Accuracy: 97% |
| M. Daradkeh et al., 2022 [26] | Scientometrics | CNNs Models | Unknown | Accuracy: 81% |
| D. Yu et al., 2023 [40] | Sentiment Classification of Scientific Citation | BERT, SCIBERT, DictSentiBERT, LSTM, FNN, TextCNN, Self-Attention | Resources [12,13] | DictSentiBERT (BERT) Accuracy: 93.49%, DictSentiBERT (SCIBERT) Accuracy: 95.20% |
| N. Yang et al., 2023 [42] | Entity Citation Recognition, Binary Classification | Random Forest, Decision Tree, TextCNN, TextRCNN, BERT, SCIBERT, BIOBERT | PubMed Central (PMC) | SCIBERT and BIOBERT F1-score: 85.30% |

[1] Resources by Athar: "https://cl.awaisathar.com/citation-sentiment-corpus/ (accessed on 8 April 2024)"; [2] Resources by Tsakalidis (github): "https://github.com/socialsensor/sentiment-analysis/tree/master/src/main/resources (accessed on 10 April 2024)"; [3] Palo Digital Technologies Ltd. (Athens, Greece): "https://www.palo.gr/ (accessed on 10 April 2024)"; [4] OpinionBuster has been developed as part of the Ellogon Platform: "http://www.ellogon.org (accessed on 10 April 2024)"; [5] "https://www.real.gr/ (accessed on 11 April 2024)"; [6] "https://www.kathimerini.gr/ (accessed on 11 April 2024)"; [7] Yelp 2013: "https://www.yelp.com/dataset/documentation/main (accessed on 11 April 2024)"; [8] IMDb Movie Reviews: "https://paperswithcode.com/dataset/imdb-movie-reviews (accessed on 12 April 2024)"; [9] "https://www.kaggle.com/ (accessed on 12 April 2024)"; [10] Traveloka website: "https://www.traveloka.com/ (accessed on 13 April 2024)"; [11] Resources by Billianos (github): "https://github.com/DimitrisBil/greek-sentiment-analysis (accessed on 13 April 2024)"; [12] (github): "https://github.com/UFOdestiny/DictSentiBERT (accessed on 13 April 2024)"; [13] (github): "https://github.com/allenai/scibert/tree/master/data/text_classification/sci-cite (accessed on 14 April 2024)".

## 5. Discussion

This section will answer the Research Questions noted in Section 2.1.

- *RQ1.* Machine Learning based techniques, such as SVM, Naive Bayes, and Decision Tree, and advanced Machine Learning models, such as LSTM, BERT, RoBERTa, and BioBERT, have provided significant improvements in the accuracy of detection and the analysis of emotions. Deep Learning models have shown wonderful progress because they can identify semantic patterns in the data. However, Deep Learning requires significant computational resources and expertise, while traditional methods are often simpler and more accessible.

- *RQ2.* In our review, the researchers used several preprocessing methods, such as removing unimportant words (stopwords) from the text and converting words to vectors using TF-IDF and Word2Vec techniques. Additionally, precision, recall, and accuracy were used as evaluation metrics.

- *RQ3.* Machine Learning models, such as SVM, Naive Bayes, Decision Tree, etc., may perform better in applications where data is limited or where parameters need to be slightly modified. In contrast, Deep Learning models, such as CNN, LSTM, BERT, etc., are more suitable in cases of large and complex datasets. This is confirmed in [15], where a small dataset was used and the SVM achieved excellent classification accuracy, coming very close to the BERT model.

- *RQ4.* In Sentiment Analysis, and classification tasks in general, the two main types of learning used are Supervised Learning and Unsupervised Learning. Supervised Learning is particularly popular because of its ability to provide accurate predictions based on labelled data, which is critical in Sentiment Analysis. Unsupervised Learning is a type of Machine Learning where models are trained on previously unlabeled data.

Its goal is to discover hidden patterns in the data. In our review, we observed the implementation of Supervised Learning.

- *RQ5.* Sentiment Analysis allows for the identification of both positive and negative emotions in scientific citations, increasing the ability to critique and understand the motivations behind scientific findings. By understanding the emotion conveyed through scientific texts, researchers can improve communication and collaboration among themselves. Recognizing the emotional cues in texts can help avoid misinterpretations and create more constructive communication.

- *RQ6.* Challenges include dealing with complex scientific terminology, multilingualism, and the abstract nature of discussions that require specialized language processing techniques.

- *RQ7.* In addition to polarity detection, many researchers, as we observed in our review, apply classification based on the purpose of the citation. For example, a frame of reference can be supportive (supportive type) and reinforce an idea or viewpoint presented in the text, critique another research (critique type), be used to compare research results of papers (comparison type), document important previous studies that support or influence the current research (documentation type), or even refer to a paper that forms the theoretical background of the current study (base type).

- *RQ8.* The availability of public datasets is still limited. Although there are some sources that offer access to scientific articles and their references, datasets that include labeled citation contexts are rare. One reason for this relates to the copyright that protects scientific documents. Moreover, in the case of Supervised Learning it is necessary to label citations manually, which makes it a complex process.

- *RQ9.* Emotions play a crucial role in communicating scientific results, as they influence the acceptance of information by the scientific community and the wider public. Emotions can strengthen or weaken the persuasiveness of arguments, and they can also encourage confidence in findings or, conversely, cause doubt. For example, a scientific article that receives more positive citations may stimulate more interest and active acceptance, while an article that receives negative citations may potentially raise reservations among other researchers.

## 6. Future Research

Some recommendations for improvement in future related work are as follows:

- *Increase data.* By increasing the amount of data, models become more accurate and achieve higher generalization. In addition, the ability to collect data from different platforms offers a more comprehensive approach to analyzing emotions.

- *Combination of different types of data.* Merging information, such as text, image, audio, and video, can improve the accuracy and completeness of sentiment analysis.

- *Pre-process methods.* Data processing prior to model training can have a major impact on the final performance. The choice of the most appropriate pre-processing method depends on the nature of the data and the goal of each application.

- *Model selection.* The process of selecting the appropriate model for solving a Machine Learning problem is also a very important process. Any model trained on specific data will perform well on such new data.

- *Architecture.* The use of more complex Neural Network architectures (number of layers and neurons) clearly affects the performance of the models.

- *Analysis of implicit and explicit citations.* Extensive studying of the distinction and interpretation of implicit and explicit citations within scientific texts for a better understanding of purpose and polarity.

- *Citation context retrieval methods.* Focus on developing and improving methods for retrieving, processing, and analyzing the citation context, including more advanced approaches to reveal its deeper meaning.

Having reviewed the current challenges in the field of research regarding the analysis of polarity in scientific texts, it is important to mention the prospects for future work. In the

next stage of research, emphasis will be placed on the development of NLP and Machine Learning methods. An important goal is to create a new dataset for both experimentation and detecting polarity in scientific publications, as well as for comparing the results with those reported by the research studies reviewed in this paper. Also, the intention behind a citation in a scientific article will be investigated. Finally, there is the consideration of developing a Citation Recognition System using pre-trained language models based on BERT.

## 7. Conclusions

This research approached the analysis of emotions in text and scientific publications by combining techniques from the fields of Machine Learning and Deep Learning, highlighting the need for more advanced methods for detecting and evaluating emotional nuances. Through the analysis of the polarity of emotions and understanding the purpose of citations, their complexity and importance in scientific communication was revealed. With the help of the research papers reviewed, this study highlighted the need for further research and development in this area, enhancing the understanding of the value and influence of scientific papers.

**Author Contributions:** All authors have contributed to the review presented. Conceptualization, A.K. and A.S.; writing original draft preparation, A.K. and A.S.; writing review and editing A.K., A.S., K.D. and S.O.; visualization, A.K.; supervision, A.S., K.D. and S.O.; resources, A.S., K.D. and S.O.; All authors have read and agreed to the published version of the manuscript.

**Funding:** This review article has not received external funding.

**Data Availability Statement:** Data are contained within the review article.

**Conflicts of Interest:** The authors declare no conflicts of interest.

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
