# Peer review of "Sentiment Dimensions and Intentions in Scientific Analysis: Multilevel Classification in Text and Citations"

_electronics, doi:10.3390/electronics13091753_

Round 1
Reviewer 1 Report
Comments and Suggestions for Authors
This paper reviews textual sentiment analysis on scientific texts and citations. The PRISMA methodology is used to establish a proper approach, resulting in a four-step systematic reviews. In general, I do not have any main concern of this review. Instead, there are some suggestions that authors may consider:
1. Add more details of search queries in Table 1, for example, how these queries are determined, and why different databases have various queries (e.g,. why IEEE Xplore has only one query.)
2. Figures and charts in this paper show insufficient resolution, please consider replacing them with vector diagram.
3. I suggest including a subsection on the preprocessing steps necessary for handling scientific texts.
4. The challenges section is insightful, but it could benefit from discussing specific examples where these challenges have significantly impacted research outcomes.
5. It would be beneficial to include more recent studies, such as large language models like ChatGPT/GPT-4.
Author Response
Author's Notes
We would like to thank the reviewer for the time they dedicated to review our manuscript. We believe that by addressing the reviewer’s comments, the overall quality of our manuscript has increased. We hope that the comments and changes made will be assessed as satisfactory.
Below, we quote our responses to each of the comments made:
Comment:
Add more details of search queries in Table 1, for example, how these queries are determined, and why different databases have various queries (e.g., why IEEE Xplore has only one query.).
Response:
Search queries are tailored to provide specific results based on the characteristics of each database. This led us to different search queries depending on the need for precision or broad coverage of results. Added text to 2.2 Search strategy and selection criteria.
Comment:
Figures and charts in this paper show insufficient resolution, please consider replacing them with vector diagram.
Response:
The same images were created again.
Comment:
I suggest including a subsection on the preprocessing steps necessary for handling scientific texts.
Response:
Added Section 3.3.2. Text and Citation Preprocessing.
Comment:
The challenges section is insightful, but it could benefit from discussing specific examples where these challenges have significantly impacted research outcomes.
Response:
We particularly appreciate the point about adding concrete examples showing the impact of challenges on research results. However, our intention was to present a general overview of challenges in sentiment analysis. Due to the general nature of the work and the lack of personal experimental data, we did not include examples, but we did proceed to update the challenges section.
Comment:
It would be beneficial to include more recent studies, such as large language models like ChatGPT/GPT-4.
Response:
We agree with the reviewer that it would be good practice to include studies with language models such as ChatGPT/GPT-4. In our review we mentioned models such as BERT, RoBERTa, SciBERT etc., which are also important models. An analysis of recent studies and technologies would require further research and time, which was not available within the scope of the current work. However, we will consider the reviewer's suggestion for future research work.
Reviewer 2 Report
Comments and Suggestions for Authors
The justification and originality of the study are clear and relevant, however the research problem and the objective of the article were not clearly presented in the introduction.
In the methodology the authors present 11 research questions. It is suggested that the authors readjust the questions and present the general focus of the article.
Not all of the research questions have been answered, but there is another reason for them to be grouped together.
The methodology was adequate and well structured.
The conclusion of the study was done in a clean manner. It is necessary to better answer the questões that have been answered.
A section is suggested for practical implications and for academia.
Author Response
Author's Notes
We would like to thank the reviewer for the time they dedicated to review our manuscript. We believe that by addressing the reviewer’s comments, the overall quality of our manuscript has increased. We hope that the comments and changes made will be assessed as satisfactory.
Below, we quote our responses to each of the comments made:
Comment:
The justification and originality of the study are clear and relevant, however the research problem and the objective of the article were not clearly presented in the introduction.
Response:
We agree with the reviewer. added a paragraph to the introduction.
Comment:
In the methodology the authors present 11 research questions. It is suggested that the authors readjust the questions and present the general focus of the article.
Response:
We agree with the reviewer. The research questions have been adjusted.
Comment:
Not all of the research questions have been answered, but there is another reason for them to be grouped together.
Response:
We agree with the reviewer. A "Discussion" section was created in which the research questions we defined in section 2 are answered.
Comment:
The methodology was adequate and well structured.
Response:
Thank you very much for the comment on the methodology of the survey. It is encouraging to know that this effort is recognized and appreciated by the jury.
Comment:
The conclusion of the study was done in a clean manner. It is necessary to better answer the questões that have been answered.
Response:
We agree with the reviewer. We have improved the research questions. We have grouped them and answered them in the "Discussion" section.
Comment:
A section is suggested for practical implications and for academia.
Response:
Thank you for your suggestion to include a chapter on practical applications and implications for the academic community in our research. We recognize the importance of such an addition to enrich the discussion. Due to time constraints, this time it was not possible to include this section in the current version of the article. However, we will consider the possibility of including this addition in a future review, as we believe it will provide great value.
Reviewer 3 Report
Comments and Suggestions for Authors
This paper presented systematic literature review to investigate the application of Machine and Deep Learning models in the sentiment analysis field. The research topic is interesting and within the scope of the journal. However, the authors should address the following major and minor comments in their manuscript to increase the quality of the paper.
Major comments
1. The main question addressed by this review article is to trace the application of Machine and Deep Learning models in emotions analysis in scientific publications. The focus of the review article was on the application of Machine and Deep Learning models in overcoming challenges faced by conventional approaches.
2. The review article presented a deep understanding of the evolution of the integration of Machine and Deep Learning models in the sentiment analysis. In particular, the importance and complexity of emotion polarity analysis of scientific citation was discovered. In addition, the results of this review article emphasis the requirement of learning models for effective knowledge of the importance and effect of scientific papers.
3. In the introduction section, please emphasis more on the originality of the paper. For example, you can cite the previous conducted literature review papers in the field and then conduct a comparison between them and the proposed systematic literature review.
4. In section2, the time span of the selection and rejection criterion is not defined in the methodology.
5. In Line 186: please elaborate more about the reasons of the selection of papers from the references of other articles or websites.
6. In table 8, the authors presented a comparison between the selected papers, but only 20 studies are included. Please include all the selected papers in the table.
7. Section 3 presented a summary of the major findings of the study. However, the 11 research questions posed by the authors were not clearly addressed by the authors. It is highly recommended to add a discussion section to discuss and answer in details research questions.
8. The authors included 37 studies in the systematic literature review, but they did not cite all the 37 studies in the references. It is highly recommended to include all the studies in the references list to be easily accessed by readers.
Minor comments:
1. The title of figures 3 and 5 should be changed to clearly represent the content of the figure.
2. The quality of figures 2,3,4,5 should be improved. In addition, the font size is too small in these figures to be read.
3. The quality of figure 7 should be improved.
4. The caption of table 8 includes too many references, they can be better added to references list not in caption.
Comments on the Quality of English Language
Minor editing of English language required.
Author Response
Author's Notes
We would like to thank the reviewer for the time they dedicated to review our manuscript. We believe that by addressing the reviewer’s comments, the overall quality of our manuscript has increased. We hope that the comments and changes made will be assessed as satisfactory.
Below, we quote our responses to each of the comments made:
Comment:
The main question addressed by this review article is to trace the application of Machine and Deep Learning models in emotions analysis in scientific publications. The focus of the review article was on the application of Machine and Deep Learning models in overcoming challenges faced by conventional approaches.
Response:
Thank you for your comment and for identifying the main question of the review, which focuses on the application of Machine Learning and Deep Learning models to sentiment analysis, as well as addressing the challenges encountered. It is encouraging to see that the content of the paper has been well understood and appreciated by the reviewers.
Comment:
The review article presented a deep understanding of the evolution of the integration of Machine and Deep Learning models in the sentiment analysis. In particular, the importance and complexity of emotion polarity analysis of scientific citation was discovered. In addition, the results of this review article emphasis the requirement of learning models for effective knowledge of the importance and effect of scientific papers.
Response:
Thank you very much for your encouraging words and for acknowledging the deep understanding our article presented about the evolution and integration of Machine and Deep Learning models in Sentiment Analysis. This recognition reinforces our commitment to continue to research and deepen our understanding of this important field, seeking to provide even more practical applications in the future. We appreciate the time and attention you have taken to analyze and evaluate our work.
Comment:
In the introduction section, please emphasis more on the originality of the paper. For example, you can cite the previous conducted literature review papers in the field and then conduct a comparison between them and the proposed systematic literature review.
Response:
We agree with the reviewer. Added a paragraph in the Introduction and a paragraph in section 2. Research Methodology.
Comment:
In section2, the time span of the selection and rejection criterion is not defined in the methodology.
Response:
We agree with the reviewer. Updated the text in 2.2 Search Strategy and Selection Criteria. In general, we did not apply strict temporal search filters.
Comment:
In Line 186: please elaborate more about the reasons of the selection of papers from the references of other articles or websites.
Response:
We agree with the reviewer. Added text before Table 1.
Comment:
In table 8, the authors presented a comparison between the selected papers, but only 20 studies are included. Please include all the selected papers in the table.
Response:
At this point we should explain that, to make a comparison, only those studies that were examined in depth and provide experimental results are included in Table 8. Studies that we simply took text only to enhance our review are not included in the table.
Comment:
Section 3 presented a summary of the major findings of the study. However, the 11 research questions posed by the authors were not clearly addressed by the authors. It is highly recommended to add a discussion section to discuss and answer in details research questions.
Response:
We agree with the reviewer. A "Discussion" section was created in which the research questions we defined in section 2 are answered.
Comment:
The authors included 37 studies in the systematic literature review, but they did not cite all the 37 studies in the references. It is highly recommended to include all the studies in the references list to be easily accessed by readers.
Response:
All studies have been included in the reference list. In fact, we have 17 papers (Journal Article) and 20 Conference Papers found in digital repositories. In addition, we have 4 Websites and 4 papers (Journal Article) and 1 Conference Paper found in the above 37 papers retrieved from Digital Repositories and included in the review. Finally, 1 paper is related to the PRISMA methodology. Total 47 references.
Comment:
The title of figures 3 and 5 should be changed to clearly represent the content of the figure.
Response:
The title of the pictures has been changed.
Comment:
The quality of figures 2,3,4,5 should be improved. In addition, the font size is too small in these figures to be read.
Response:
Τhe quality of the images has improved. The font size has been increased.
Comment:
The quality of figure 7 should be improved.
Response:
The quality of picture 7 has been improved.
Comment:
The caption of table 8 includes too many references, they can be better added to references list not in caption.
Response:
We understand your urging to simplify the caption by moving the references to the list of references. The choice to include these references directly in the footer of the table was made to provide the reader with direct access to the references that are most relevant to the specific data and resources presented. This helps to better understand the content of the table and enhances the coherence of the data presentation, avoiding the need to continually go through the list of references.
Round 2
Reviewer 2 Report
Comments and Suggestions for Authors
All requests were met. The manuscript was revised and substantially improved, which is why I suggest approval of the submission.